# Investigating the Electromechanical Sensitivity of Carbon-Nanotube-Coated Microfibers

**DOI:** 10.3390/s23115190

**Published:** 2023-05-30

**Authors:** Elizabeth Bellott, Yushan Li, Connor Gunter, Scott Kovaleski, Matthew R. Maschmann

**Affiliations:** 1Department of Mechanical & Aerospace Engineering, University of Missouri, Columbia, MO 65211, USA; 2Department of Electrical Engineering & Computer Science, University of Missouri, Columbia, MO 65211, USA; 3MU Materials Science and Engineering Institute, University of Missouri, Columbia, MO 65211, USA

**Keywords:** piezoresistance, carbon nanotube, microfiber

## Abstract

The piezoresistance of carbon nanotube (CNT)-coated microfibers is examined using diametric compression. Diverse CNT forest morphologies were studied by changing the CNT length, diameter, and areal density via synthesis time and fiber surface treatment prior to CNT synthesis. Large-diameter (30–60 nm) and relatively low-density CNTs were synthesized on as-received glass fibers. Small-diameter (5–30 nm) and-high density CNTs were synthesized on glass fibers coated with 10 nm of alumina. The CNT length was controlled by adjusting synthesis time. Electromechanical compression was performed by measuring the electrical resistance in the axial direction during diametric compression. Gauge factors exceeding three were measured for small-diameter (<25 μm) coated fibers, corresponding to as much as 35% resistance change per micrometer of compression. The gauge factor for high-density, small-diameter CNT forests was generally greater than those for low-density, large-diameter forests. A finite element simulation shows that the piezoresistive response originates from both the contact resistance and intrinsic resistance of the forest itself. The change in contact and intrinsic resistance are balanced for relatively short CNT forests, while the response is dominated by CNT electrode contact resistance for taller CNT forests. These results are expected to guide the design of piezoresistive flow and tactile sensors.

## 1. Introduction

In nature, hair cells and hairlike sensors (HLS) are crucial to the survival of many species, providing a means to identify and react to environmental stimuli and dangers. Biological hair cells have been found to be highly sensitive and efficient [1]. For example, celia hair arrays on crickets can detect small airflow disturbances from approaching predators [2]. Likewise, specialized hairs on the wings of bat species inform navigation and assist with flight control [3]. Many kinds of artificial HLS have been fabricated to mimic such structures and their functions [4]. However, current artificial HLS are inferior in performance to their biological counterparts, which rely on neurons and neuronic membranes for signal processing [5].

HLS designs incorporate a hair component that receives a signal and a pore structure that processes or converts the signal. HLS have been designed to capture a variety of signal types representing different environmental stimuli related to flow [6,7,8], touch [9,10] vibration [11], chemistry [12], and acceleration [13]. The conversion of piezoresistive [12,14], piezoelectric [15,16], capacitive [8], or magnetic [17,18] signals into electrical data can provide magnitude and directional information related to external stimuli. In other words, the mechanical deformation of a hair structure imparts strain or resistance at the pore structure which is converted to an electrical signal. The performance of HLS is determined by signal sensitivity, or detection of low magnitude stimuli, and the range of signal detection which is dictated by material, geometric, and size constraints. To maximize these measures of performance, previous HLS designs have implemented various material properties found with polymers [19], semiconductors [5], metals [20], and oxides [21]. Various processing methods including microelectromechanical systems (MEMs), related technology, and micromachining have been used to accommodate such different materials, fabricate sensors at the micro- and millimeter scale, and produce arrays of sensors [1,4,5,6,7].

Carbon nanotube (CNT) forests are vertically oriented films that may be synthesized at the microscale or to dimensions exceeding 1 cm [22]. CNT forests’ compressive mechanical properties have been examined extensively using indentation methods [23,24,25,26,27,28,29,30,31,32,33]. The typical compressive modulus is on the order of 1–100 MPa, depending on the synthesis conditions. The effective modulus of a CNT forest may be increased by adding a thin atomic layer deposition coating [34,35] and capillary densification [36] or decreased via fluid immersion [37]. The electromechanical response of CNT forests has been investigated to a lesser extend [38,39]; however, the interconnected, open-cell foam structure of CNT forests make them candidates for a sensitive piezoresistive response. The interconnected morphology, high durability [40], and conductive properties of CNT forests generate a large piezoresistive response that make them candidates for compliant flow or proximity sensors. 

CNT-forest-coated microfibers were previously employed as an HLS to detect low-velocity air flow [38]. In this construction, the CNT-coated fiber was positioned within a pore structure that contained a set of electrodes spaced along the axial direction of the fiber. The fiber extended through the length of the pore and above the pore into the airflow for several millimeters. The deflection of the exposed fiber induced by airflow locally compressed the CNT forest coating along the wall of the pore, and a decrease in electrical resistance was detected along the length of the pore. The CNT forest transducer produced a signal that could detect airflow up to 1 m/s and deflection frequencies of at least 10 Hz. The deflection of the exposed hair tip was on the order of 1–10 μm. The CNT-coated fiber outer diameter was on the order of 25 μm for these demonstrations. An electromechanical study of isolated CNT-forest-coated microfibers of various CNT morphology and coating diameter has not previously been conducted to determine the optimal CNT coating parameters for piezoresistive transduction.

Herein, we examine the piezoresistive response of compressed CNT-forest-coated microfibers using experiments and simulation. The CNT forests are synthesized on 8–10 μm diameter glass fiber substrates to final diameters greater than 100 μm using floating catalyst chemical vapor deposition. The internal morphology of CNTs is varied by adding a thin alumina coating to the glass substrate, resulting in decreased CNT diameter and increased areal density. The coated fibers are then compressed against patterned electrodes to determine their piezoelectric response. Compressive gauge factors greater than three are observed, with 35% resistance reduction per micron of compression. The synthesis and electromechanical compression of CNT-forest-coated fibers is also examined using a finite element simulation. The simulated results provide insights into the relative contributions of CNT–CNT contact resistance and CNT–surface contact resistance that govern the piezoresistive sensitivity in the percolative transport system. 

## 2. Materials and Methods

### 2.1. CNT Synthesis of Fiber Substrates

Glass microfibers (AGY 933 S-2, with a softening temperature of 1056 °C) served as the bulk substrate for all experiments. The glass microfibers were utilized in their native, as-received state or were coated nominally with 10 nm of alumina via atomic layer deposition (ALD) at 150 °C using 100 pulses of water vapor and Trimethylaluminum [41]. ALD was used to coat the fibers, because the technique is a gas-phase reaction coating mechanism. Since the technique is not line-of-sight, such as sputtering, a bundle of fibers may be conformally coated in a single deposition. Alumina is known to discourage Ostwald ripening of Fe catalyst particles, thereby reducing the CNT diameter and increasing the CNT number density [42]. The glass microfibers were nominally 8–10 μm in diameter and were cut to lengths of 2–3 mm.

CNT synthesis on the fibers occurred using atmospheric pressure floating catalyst chemical vapor deposition (CVD) method within a 20 mm diameter tube furnace. A solution of 5 weight percent ferrocene in xylenes was prepared for injection into the tube furnace via a programmable syringe pump. Prior to injection of the solution, the tube furnace was heated to 750 °C using 50 sccm H_2_ (Airgas UHP) and 500 sccm Ar (Airgas UHP) at a ramp rate of approximately 50 °C/min. The ferrocene and xylenes solution was injected into the furnace at a rate of 1 mL/hr using the same flow of gases used during the temperature ramp. Synthesis time was adjusted to tune the length of CNTs with a maximum synthesis time of 1 h. Upon synthesis completion, the injection of the ferrocene/xylenes solution was ceased, and the furnace was cooled to below 100 °C before removing the CNT-coated fibers. A schematic of the CVD setup, and the synthesis conditions used for synthesis, are displayed in Figure 1.

### 2.2. Electromechanical Compression

An Agilent G200 nanoindenter was employed for the fiber indentation. A 100 μm wide diamond flat indenter tip was used to compress the CNTs on the fiber sample. This tip geometry was chosen due to its relatively large surface area and the uniform contact geometry. A maximum load of 10 mN was applied to each fiber. The mechanical test consisted of a single loading and unloading segment. The loading and unloading segment was 15 s in duration. A schematic of the setup is displayed in Figure 2. 

Single fibers were placed on interdigitated electrodes for each test. The width of the electrodes was 5 μm with a 10 μm pitch from the midline of one electrode finger to its nearest neighbor. The electrical resistance of the CNT-coated fibers was recorded during the indentation using a Keithley 2701 model running KickStart data collection software. For each nanoindentation test, the displacement, force, and electrical resistance of the host microfiber were simultaneously recorded to provide a coupled electromechanical response. Compression was applied diametrically, while the electrical resistance was measured axially, consistent with previous electromechanical CNT sensors [38]. Care was taken to compress regions of fibers that appeared homogeneous within the nanoindenter microscope to minimize the effects of irregular forest geometries.

### 2.3. Simulation Details

The electromechanical compression of CNT-coated glass fibers was simulated using a custom finite element simulation in MATLAB in which each CNT comprises interconnected frame elements [43,44,45,46]. The simulation began with the synthesis of the CNT forest. The CNT forests were grown around an 8 μm diameter cylindrical fiber to examine evolving forest morphology. A span of 10 μm was simulated to model electromechanical compression, representing the pitch of the physical measurement electrodes. To simulate the difference between the as-received glass surface and alumina-coated surface, the diameter and areal density of CNTs were varied. Simulated CNTs synthesized from as-received glass fibers featured an outer diameter of 40 nm with an areal density between 0.5–1 × 10^9^ CNT/cm^2^. Simulated CNTs synthesized from alumina-coated fibers featured an outer diameter of 10 nm with an areal density between 1–3 × 10^10^ CNT/cm^2^.

In the synthesis phase of simulation, CNT growth occurred by adding a new finite element to the base of each CNT at discrete time steps [43,44,45,46]. This process represents the base-growth CNT growth mechanism [47,48] which is observed with floating catalyst synthesis. The CNTs within the populations were assigned growth rates consistent with a Gaussian distribution with a mean of 60 nm per time step, and a coefficient of variation of 5%. CNT–CNT contact adhesion was simulated by introducing a linear-elastic element between nodes in contact. Contacting CNTs growing at different rates and different directions generate reaction forces that bend and deform the contacting CNTs about their contact points. 

Compression of CNT-coated fibers used the same finite element framework. Compression of the forest occurred by translating a rigid platen. The platen was aligned with the long axis of the cylinder and translated normal to the fiber diameter. In these simulations, the platen moved 20 nm per discrete time step. The sum of reaction forces between CNTs and the moving platen represented the compressive force. A rigid surface extended along the bottom surface of the fiber, parallel to the compression platen. Flat electrodes were placed at the bottom rigid surface at a pitch of 10 μm and a width of 5 μm. The simulation domain in the axial direction was 10 μm. Electrodes aligned with the leading and trailing 2.5 μm of the simulation domain, consistent with physical experiments. Half of each electrode was assumed to occupy similar unit cells. CNT nodes in contact with the electrodes could conduct current within the network. The voltage difference applied to the electrodes was 10 V in our experiments. The electrical conductance of each CNT element was equal to 10^5^ S/m [49]. The electrical resistance through each CNT–CNT contact was 50 kΩ [50], and CNT–metal contact resistance was 10 kΩ [51]. The electrical resistance between measurement electrodes was obtained by dividing the potential difference between the measurement electrodes (10 V) by the current leaving the source electrode. Since no single CNT is expected to bridge between the two electrodes, the electrical resistance signal is a result of percolation resistance within the larger CNT forest network. 

## 3. Results

CNT forests were grown on bundles of several thousand glass fibers in each synthesis experiment. A fraction of the fibers was selected for SEM image analysis using a ThermoFisher Phenom Pharos G2 field emission SEM. The CNT forests synthesized on the alumina-coated and as-received fibers have noticeable differences in morphology. Example CNT forests synthesized upon an alumina-coated fiber and as-received glass fiber, at a growth time of 30 min, are shown in Figure 3. The outer diameter of these fibers was on the order of 35 μm (Figure 3a,c). A small fraction of CNTs is significantly longer than the population average and extends beyond the common termination front of CNT growth. The CNTs grown on the alumina layer are multi-walled CNTs with typical outer diameter of approximately 5–30 nm (Figure 3b). The measured diameter average and mode measured by SEM was 29.1 and 27.1 nm, respectively. The CNT outer diameter grown from the as-received fiber are much larger (Figure 3d), with a diameter average and mode of 70.4 and 50.1 nm, respectively. The alumina support layer increases the areal density of CNTs while decreasing average CNT diameter, consistent with previous observations [42]. Small iron nanoparticles decorate the surface of the CNTs, as is typical with floating catalyst CVD synthesis. Catalyst nanoparticles observed on the fiber surface under delaminated CNTs point to base growth as the predominant CNT synthesis method.

The CNT diameter, internal CNT interconnectivity, and uniformity of a CNT forest coating are expected to be key contributors to mechanical and piezoresistive sensitivity. The conductive pathway of a CNT-forest-coated fiber can be thought of as a percolation network. Electrical current, the measurement along the long axis of the fiber, must travel between many CNTs while moving from the source to the drain. The diameter and defect density of CNTs determine their intrinsic conductivity. A uniform cylindrical CNT coating around the fiber promotes a high density of CNT contacts to the measurement electrodes. High density CNTs also promote a high density of CNT contacts to electrodes, while also encouraging a greater number of CNT–CNT contacts that form the percolation network. Larger diameter CNTs have a much greater bending stiffness compared to small diameter CNTs, as the second area moment of inertia for hollow cylinders scales with diameter to the fourth power. Due to the open cell morphology of CNT forests, the bending of CNTs is the dominant deformation mode in compression. Furthermore, because of their enhanced stiffness, larger-diameter CNTs tend to demonstrate decreased tortuosity and interconnectivity in comparison to smaller diameter CNTs. Larger diameter CNTs also experience a reduced CNT–CNT van der Waals force [52]. This combination of morphological factors make the optimization of piezoresistive properties challenging. 

As demonstrated in Figure 3a, dense CNT regions tend to separate into coordinated bands that resemble a mohawk-like structure when the CNT forest coating thickness exceeds a critical value. This morphological behavior has been observed previously from CNTs upon alumina microfibers [53]. The transition from a uniform, cylindrical CNT coating to the “mohawk” coating occurs at approximately 35 μm for alumina-coated fibers and 50 μm for as-received fibers. The effect is exaggerated as the CNT forest thickness exceeds the diameter of the fiber substrate (Figure 4). This CNT forest separation and banding behavior is the result of CNTs growing in a predominantly radial direction from the cylindrical substrate. If unconnected to neighboring CNTs, the free ends of radially growing CNTs would grow farther apart with increased growth. However, with van der Waals bonding, neighboring CNTs adhere together, and the tendency to spread apart with increased CNT length generates forces in opposition to the van der Waals bonds. The forces acting to separate contacting CNTs increase until van der Waal bonds break and segmented CNT islands form. The number of predominant bands around the fiber perimeter range from one band (the fiber profile resembled an extended teardrop with the fiber at the bottom—see Figure 4b) to four bands. As outer diameters approach 100 μm, two dominant bands are often observed—each offset by approximately 180° around the fiber perimeter. The CNT forest banding and irregular CNT forest perimeter could influence the effective stiffness of the CNT-coated fiber, the electrical contact between CNTs and the measurement electrode, and the variability of the percolation network in response to compression. When compressing all CNT fibers, areas of nearly uniform cross-sectional area were selected by using the optical microscope on the nanoindenter system.

### 3.1. Electromechanical Experimental Results

Electromechanical compression was performed on individual CNT-coated fibers resting on flat interdigitated gold electrodes patterned on a quartz substrate. Compression was imposed in a diametric orientation, while electrical resistance was measured along the long axis of the fiber. Due to the cylindrical geometry of each sample, the contact area between the fiber and electrode surface and indenter tip increased as a function of compression. The maximum anticipated contact surface area is represented by the width of the indenter tip (100 μm) and the outer CNT-coated fiber diameter (varies). However, this area estimate represents an upper bound, as full contact with the full diameter of the CNT-coated fiber will not be realized. The overestimation increases as the diameter of the CNT-coated fiber decreases. Since the general orientation of each CNT is radial with respect to the cylindrical substrate, the dominant mode of deformation within the CNT forest, in response to diametric compression, is bending. A low effective stiffness is anticipated from the CNT forest coating because of the small contact area and bending of CNTs. 

Two sets of force vs. displacement data are shown in Figure 5, representing a relatively small diameter fiber (Figure 5a) and a relatively large diameter fiber (Figure 5d). Both fiber substrates were coated with alumina before synthesis. The 12 μm diameter fiber exhibits a monotonically increasing load in response to 10 mN compression. Note that the loading slope is approximately linear from 2–10 mN of loading, potentially indicating a nearly constant contact area. Unloading follows a similar slope, with a hysteresis that is commonly observed during the compression of flat CNT forests [27]. The fiber exhibits little plastic deformation. The larger 60 μm diameter CNT forest fiber was compressed 16 μm by 10 mN of compression. A small shoulder in the force vs. displacement curve between 10–12 μm of compression is similar to the plateau region of planar CNT forests when compressed [23,27]. The plateau region indicates coordinated CNT buckling that results in large plastic deformation. The plastic deformation for the 60 μm diameter fiber is on the order of 12.5 μm out of a total 16 μm compression.

The electrical resistance of the CNT fibers was measured during the indentation tests. Typical electromechanical data are exhibited in Figure 5c,f. The general resistance profile with displacement is similar between the two fibers. An initial, extended region with a large negative slope is established at small displacements, followed by a lower limit that is approached asymptotically. The data shown in Figure 5 represent cases in which the electrical resistance was restored to initial levels upon removal of the compressive load. In other tests, the electrical resistance recovered partially. Interestingly, despite the plastic deformation observed in the mechanical data of the 60 μm compressed fiber (Figure 5e), the electrical resistance fully recovered. This phenomenon might be explained by considering that the rigid glass fiber distributes the load applied by the indenter tip axially, such that CNTs below the fiber (relative to the indenter tip) experience lower force per unit area and decreased deformation. In the case of the data in Figure 5e,f, the plastic deformation may have occurred predominantly on the top side of the fiber, while the bottom side of the fiber in contact with the electrode fully recovered. 

The electromechanical sensitivity of fibers was assessed using the gauge factor. The gauge factor relates a change in electrical resistance to a change in strain. The gauge factor is defined as the normalized change in resistance relative to the diametric strain:GF=ΔRRoΔxDo=ΔRΔxDoRo
where Δ*R*/Δ*x* is defined as the resistance change as a function of radial compression, *R_o_* is the initial resistance of the fiber, and *D_o_* is the undeformed diameter of the CNT forest-coated fiber prior to compression. The slope of Δ*R*/Δ*x* is assessed at the resistance representing the average value between the minimum and maximum observed resistance value. An alternative sensitivity metric was also examined in which the undeformed diameter was removed from the calculation of gauge factor by dividing the gauge factor by undeformed diameter. This parameter relates the normalized resistance change per unit micron of deformation without regard to the initial fiber diameter.

Figure 6 shows that, for both sensitivity metrics, smaller fiber diameters are associated with a greater piezoresistive response. The decreasing trend in gauge factor as a function of diameter occurs in spite of the fiber diameter appearing in the numerator of the gauge factor equation. Small-diameter fibers feature short CNT forests that are not as well interconnected as taller CNT forests. As such, the electrical percolation network for small diameter fibers is less developed. Small diametric deformation can greatly enhance the relative interconnectivity of the short CNT forests relative to taller CNT forests. The small diameter and greater areal density afforded by the 10 nm alumina coating slightly increases gauge factor and sensitivity relative to the larger-diameter and lower-areal-density CNT forests produced on as-received glass fibers. The slightly greater sensitivity may be a result of a significantly lower bending stiffness afforded by the small-diameter CNTs, which might encourage an increased areal contact with the measurement electrodes to decrease contact resistance.

Most fibers asymptotically approach a minimal resistance value between 150–350 Ω upon 10 mN of compression. However, the initial resistance of fibers varies from approximately 100–2500 Ω, as shown in Figure 6c. Assuming that all fibers reach a similar final resistance, the difference in resistance that appears in the gauge factor equation is dominated by the initial resistance of the fiber prior to compression. Though the data are significantly scattered, a general trend of decreasing initial resistance with increasing diameter is observed. We believe that the data scatter may arise from variations in contact resistance. With few CNT asperities making initial contact with the electrode surface, small changes from sample to sample are expected to lead to significant differences in measured resistance. This is particularly true for the smallest-diameter fibers, where high curvature is expected to further decrease the contact area between a cylinder and a flat plane. Such surface asperities are observed in Figure 3a,c. The other contributor to gauge factor metric is the compressive displacement over which the change in resistance occurs. The electromechanical relationships shown in Figure 5c,f show that the larger-diameter fiber undergoes the greater resistance change; however, since the displacement over which the resistance change occurs is much greater for the large-diameter fiber, the slope of resistance change (ΔR/Δx) is much lower. Therefore, the gauge factor and sensitivity of the smaller-diameter fiber is greater than that of the larger-diameter fiber.

### 3.2. Simulation Results

#### 3.2.1. CNT Forest Morphology

The morphology of CNT forests synthesized from a substrate with a round cross-section was simulated using a dynamic finite element simulation that included the synthesis and self-assembly of CNT forests. The simulations were conducted in 2D to simulate CNT forests with appreciable length. All simulations assumed a central fiber diameter of 8 μm. Simulations of as-received glass fiber substrates assumed an outer CNT diameter of 40 nm and inner diameter of 20 and 34 nm (50% and 85% O.D.), while simulations of alumina-coated glass fibers assumed an outer CNT diameter of 10 nm and inner diameter of 5 nm and 8.5 nm (50% and 85% O.D.). The inner-to-outer diameter ratios were assumed based on observations from prior ferrocene catalyst CNT syntheses [54,55,56,57,58,59,60]. CNTs were nucleated with a uniform spacing based upon the areal density. The growth rate and original orientation angle were assigned based on Gaussian distributions with a 5% covariance. CNT–CNT adhesion was achieved using linear elastic elements between contacting nodes. To evaluate the evolution of CNT forest morphology about a cylindrical substrate, the CNT–CNT interaction force was evaluated at each time step, and CNT–CNT contact elements were removed if the tensile force exceeded prescribed values ranging from 0.1 nN to 10 nN. The magnitude of force required to break the CNT–CNT van der Waals bonds is uncertain but is anticipated to be a strong function of contact length and inversely proportional to the CNT’s outer diameter [52].

Synthesis simulations were launched to assess the influence of CNT synthesis attributes to the banding of the CNT forest coating, as observed experimentally (Figure 5). In this study, the CNT areal density, CNT diameter, and CNT–CNT adhesion force were varied in a combinatorial manner. Simulations were run to 600 time steps, corresponding with an outer CNT forest diameter of approximately 50–60 μm. The typical outputs from the simulations are shown in Figure 7. Note that the total number of bands observed ranged from 1–4, in agreement with physical observations. One general trend is that a high CNT density, high adhesion strength, and small diameter produce 1–2 bands for the simulated time scales. These conditions coincide with those anticipated for CNT forest synthesis conducted on an alumina-coated glass fiber. Lower CNT density, increased diameter, and decreased CNT–CNT adhesion generates multiple bands, often between 3–4 bands. This type of synthesis would be most common for the CNT forest morphologies generated on the as-received glass fibers. While these trends are observed, each simulation assigns CNT attributes stochastically, so each simulation provides unique results. One set of synthesis parameters input to the simulation multiple times will yield variable results. We note that the number of bands observed is a function of CNT growth time with fewer bands observed with increased time. A time evolution of CNT forest growth shown in Figure 5 shows the reconfiguration of four apparent bands to one with increased simulation time. The simulated morphologies agree well with direct SEM observations.

#### 3.2.2. Simulated Electromechanical Compression

Using a similar construct as the previous simulation, the electromechanical compression of CNT forests was simulated. In these simulations, a 50% ratio of CNT inner-to-outer diameter ratio was maintained. Again, a 2D simulation was utilized to model a system of sufficient size to compare to experiments. In these simulations, the bottom CNT forest (below the fiber) that makes contact with the measurement electrodes was simulated. These CNTs undergo the most significant deformation within the anticipated percolation network for a physical 3D fiber. A rigid horizontal platen is translated vertically to the top surface of the CNTs, simulating the transmission of compressive force from the glass fiber. A schematic of the compression simulation is shown in Figure 8.

CNT forests were simulated to approximate CNT forests synthesized from the alumina-coated and as-received glass fibers. For the alumina-coated glass fiber, the simulated CNT forest comprises inter-CNT spacings consistent with 3 × 10^10^ CNT/cm^2^ and CNT outer diameters of 10 nm. To represent the as-received glass fiber, simulated CNT forests comprise CNT spacings consistent with 3 × 10^9^ CNT/cm^2^ and CNT outer diameters of 40 nm. The growth rate average was 60 nm per time step with a 5% growth rate coefficient of variation. The synthesis simulations for compression were run to 500 time steps with output files saved every 100 time steps. Therefore, each synthesis simulation output five unique CNT forests of different forest lengths ranging from approximately 6–29 μm. The resistance values associated with the CNT elements themselves, CNT–CNT contacts, and CNT–electrode contacts are consistent with those stated in the Materials and Methods section. 

Since the CNTs are generally aligned normal to the electrodes, current must flow in a percolation network between many CNTs. The main contributors to resistance are contact resistance between CNTs and between CNTs and the measurement electrodes. The resistance contribution of the CNTs themselves was negligible. The total resistance, contact resistance, and the intrinsic CNT electrical resistance from the two CNT forest morphologies, are shown in Figure 9. CNT forests with a height of less than 20 μm were compressed to 50% of their original height, while longer CNT forests were compressed by 10 μm. The total resistance exhibits trends similar to experimental observations, although the magnitude of simulated resistance is approximately 10 times greater. The quantitative difference in resistance is attributed to the 2D nature of the simulation.

The greatest rate of resistance change for all CNT forests occurred at the onset of compression, followed by a slowly decreasing resistance. The total resistance of all forests decreased by a factor of 5–10, with most resistance decrease occurring within the first of 1 μm compression. The resistance decreased slowly at displacements between 3–10 μm. The change in resistance at initial contact is on the order of 10–100 kΩ/μm at initial contact to 0.1–1 kΩ/μm after several microns of compression. The largest resistance change was observed for the shortest CNT forests, as they exhibited the greatest initial resistance. The resistance at large compression (5–10 μm) was similar for all CNT forest heights. The magnitudes of resistance, resistance change, and trends relative to changes in CNT forest height are similar between the low-density and high-density CNT forest morphologies.

Changes in electrical contact resistance may only occur by increasing the quantity of CNT–electrode contacts. The contact resistance for nodes in contact with an electrode is not an explicit function of pressure. The maximum possible contact resistance for the CNT forest simulations is 20 kΩ, corresponding to one CNT–electrical contact (each 10 kΩ), at each of the electrodes. All forests exhibited an initial contact resistance between 11–20 kΩ. The contact resistance of the dense CNT forest (10 nm O.D., 3 × 10^10^ CNT/cm^2^) in Figure 9b smoothly decreased with increased compression, with the greatest rate of change occurring within the first 1 μm of compression. The low-density CNT forest (40 nm O.D., 3 × 10^9^ CNT/cm^2^) in Figure 9e exhibited well-defined resistance plateaus within the initial 1.5 μm of compression, with each plateau indicating a static number of electrical contacts. The most rapid change in contact resistance for the lower density forests occurred within the first 3 μm of compression. The contact resistance plateaus were most distinct for the tallest CNT forests, indicating that compression was being accommodated by internal deformation rather than the establishment of a new contact area at the rigid surface. At extended compression, the contact resistance of the higher density forests continuously decreased, indicating the continuous establishment of new electrical contacts. By contrast, the lower density CNT forests reached a lower plateau in resistance between 3–4 kΩ. The large diameter of the 40 nm O.D. CNTs found in the lower-density CNT forests make each CNT more rigid and, correspondingly, decrease the mechanical compliance of the CNT forests’ free surface. 

The internal resistance was calculated by setting the voltage of each CNT node contacting an electrode equal to the voltage of the electrode itself. In this calculation, the intrinsic resistance is a function of CNT–electrode contacts in addition to the number of internal CNT–CNT contacts. The shortest CNT forests exhibited a rapid change in internal resistance in compression that was similar to, or greater than, that observed for the contact resistance. For both forest densities, the internal resistance of the shortest CNT forest (6 μm) decreased by approximately 15 kΩ within the first 1 μm of compression. This rate of resistance change is greater than the corresponding change in contact resistance. The initial internal resistance of CNT forests decreased with increased forest length in all instances, indicating a greater quantity of CNT–CNT contacts. The magnitude of internal resistance for all CNT forest heights converged at approximately 6–7 μm compression for both CNT forest morphologies and then slowly decreased with increased compression. The high-density forest exhibited an internal resistance of approximately 1 kΩ at maximum compression, while the low-density forest exhibited an internal resistance of approximately 2 kΩ.

Since the mechanisms governing the current flow within the CNT forest are CNT–CNT contacts and CNT–electrode contacts, the electrical conductance is plotted as a function of these quantities in Figure 10. The conductance is plotted instead of resistance because conductance is directly proportional to increases in current flow between electrodes. The corresponding plots with respect to electrical resistance may be found in Appendix A. The conductance displayed in Figure 10 is the total conductance (the sum of the contact and intrinsic conductance). The conductance as a function of CNT–electrode contact points is displayed in Figure 10a,b for the high-density and low-density forests, respectively. For both morphologies, the conductance increased nearly linearly as a function of CNT–electrode contact points, indicating a strong correlation. Typically, the conductance at any given number of CNT–electrode contacts does not increase appreciably until a new CNT–electrode contact is established. However, the low-density, 40 nm O.D. CNT forests exhibited appreciable conductance increases in a regime between 10–15 CNT–electrode contacts without adding additional CNT–electrode contacts. In this regime, conductance increased solely due to increasing CNT–CNT contacts. 

The conductance as a function of internal CNT–CNT contacts is shown in Figure 10c,d. The initial conductance traces for each CNT’s height are well separated because of the increased number of CNT–CNT contacts with increased CNT forest height. For all CNT forests, an initial drastic increase in conductance is observed with the addition of relatively few CNT–CNT contacts. This conductance increase is related to increases in CNT–electrode contacts rather than CNT–CNT contacts. After the initial rapid conductance increase, an extended linear relationship between conductance and CNT–CNT contacts developed, indicating a strong correlation between conductance and the formation of new internal CNT–CNT contacts. Note that the quantity of CNT–CNT contacts was on the order of tens or hundreds of thousands, while the CNT–electrode contacts ranged between 0–19 for the low-density, 40 nm O.D. CNT forest, to 0–193 for the high-density, 10 nm O.D. CNT forest. When examining the high-density forest, the extended linear relationship in conductance with respect to both CNT–electrode contacts and CNT–CNT contacts suggests that both mechanisms contribute to the increased conductance after the initial dominant contribution by CNT–electrode contacts. The abrupt step changes in conductance vs. CNT–CNT contacts exhibited by the lower-density CNT forest indicate the establishment of new CNT–electrode contacts.

A plot showing the relationship between CNT–CNT contacts and CNT–electrode contact evolution with compressive displacement is shown in Figure 11. For both CNT forest densities, the quantity of CNT–CNT contacts evolves smoothly with compression. The number of CNT–CNT contacts increases nearly linearly throughout compression for the low-density CNT forest (Figure 11a). By contrast, the high-density CNT forests exhibit an extended linear regime followed by an increase in CNT–CNT contact rate near the end of compression indicative of CNT forest densification. The increased rate of CNT–CNT contacts is accompanied by an increased rate of CNT–electrode contacts with compression within the densification regime. The quantity of CNT–electrode contacts rapidly increases, followed by extended plateaus of slowly increasing contacts. The initial rapid increase occurs because many CNT free tips are located near (but not in contact with) the electrodes at initial compression. The interface between the CNT forest and electrode surface is supported by few CNT contacts which readily deform at low loads. Once additional CNT contacts are established, and the load is distributed to all contacting CNTs, the rate of establishing new CNT–electrode contacts diminishes. Upon CNT forest densification, the internal cellular structure of a CNT forest collapses, and initially separated CNTs make contact, producing additional CNT–CNT contacts. Since the cellular nature of the CNT forest collapses, little additional deformation can occur within the internal CNT forest structure, and deformation is accommodated at the CNT–electrode interface. This effect produces a secondary increase in the rate of CNT–electrode contacts observed for many of the CNT forests in Figure 11.

## 4. Discussion

The piezoresistance of CNT forests is a complex function of the CNT forest density, the CNT diameter within the forest, and the uniformity of the surface contacting the measurement electrodes. The greatest driver of sensitivity appears to be the formation of new electrical contacts between the free ends of the CNT-coated fiber and the measurement electrodes. This agrees with previous van der Pauw measurements obtained from planar CNT forests [61]. The mechanics of CNT forests ensure that small loads induce sufficient mechanical deformation of CNTs against the rigid electrode substrate to establish many new CNT-electrode contacts, and the deformation at the free ends is recoverable upon removal of load. The small contact area established between a CNT and the electrode surface generates a relatively high contact resistance that is highly sensitive to the number of CNT contact points. The intrinsic internal resistance change established by varying the number of CNT–CNT contacts is secondary in magnitude to the establishment of new contacts with the electrode. The large and repeatable change in electrical resistance, particularly for short CNT forests, suggests that CNT forests are robust piezoresistive sensors for low-loading applications, such as artificial hairlike sensors or tactile sensors.

The shortest CNT forests represent a unique scenario. The CNT–CNT interconnectivity is not well developed for short CNT forests, leading to an increased internal resistance prior to deformation. A small deformation is sufficient to induce both a rapid increase in CNT–electrode contacts and a proportionally large increase in CNT–CNT contacts. For example, the tallest low-density CNT forest (29 μm tall) begins with 23,900 CNT–CNT contacts before compression and terminates with 37,435 contacts after 10 μm compression, representing an increase of 57%. The shortest forest (6 μm) begins with 2959 CNT–CNT contacts and ends with 8390 contacts, representing an increase of 184% over a compression range of just 3 μm. Likewise, the 6 μm generates 16 CNT–electrode contacts after 3 μm of compression, while the 29 μm CNT forest establishes only 12 over 10 μm of compression. Similar trends were observed for the high-density CNT forests. Simulations and experiments also demonstrate that the shortest CNT forests (smallest diameter fibers) exhibit the greatest initial resistance prior to compression, thereby facilitating the greatest potential change in resistance. Based on in situ SEM compression, short CNT forests also have the tendency to deform elastically under compression and recover nearly fully upon removal of the load, whereas longer CNT forests develop permanent accordion-style buckles [24]. These factors motivate the use of short CNT forests for piezoresistive mechanical sensors.

An additional consideration must be provided to the definition provided for the intrinsic internal resistance of CNT forests. The internal resistance was computed based upon applying a known potential to all nodes in contact with electrodes. As such, the internal resistance decreases (conductance increases) appreciably as new CNT–electrode contacts were established. This influence is clearly observed by the irregular relationships between the conductance and the quantity of CNT –CNT contacts in Figure 10c,d, and the nearly linear relationship between the conductance and the CNT–electrode contacts in Figure 10a,b. These results emphasize that the dominant mechanism contributing to the piezoresistance of the compressed CNT forests is the establishment of CNT–electrode contacts.

While the current 2D simulation methodology provides qualitative insights into the synthesis and electromechanical compression of CNT forests, the 2D assumptions prevent full quantitative validation of experimental results. Future simulations will be run using a 3D model based on the same principles as those used in the current study. A 3D simulation would relax many of the assumptions and estimations necessary for a 2D simulation. For example, the morphology of a 3D cylinder may be modeled and compared directly to physical CNT-forest-coated glass fibers. The effect of 2D increases in surface area between CNT forest free ends, and the measurement electrodes could also be considered in addition to the influence of banded morphology during compression. The current simulations are considered as qualitative surrogates of a 3D simulation, and the simulated results may not be compared directly with physical data for this reason. Nevertheless, the 2D simulation provides insights into both the evolution of CNT forest morphology grown from curved surfaces and the relative role of CNT–CNT contact resistance, and CNT–electrode contact resistance, during percolation-based electrical transport within CNT forests. The inherent electrical resistance of individual CNTs is negligible in comparison to contact resistance contributions. 

The response of individual CNT-coated microfibers is highly variable, even within populations of fibers synthesized at the same time. The large variation is likely a byproduct of highly variable surface properties, as has been observed by SEM. A machine learning (ML) algorithm that can learn the response of each individual piezoresistive transducer element would be ideally suited to predict the response of each CNT-coated microfiber. The ML algorithm could learn the response of a given sensor as a function of orientation and sensor deflection. Our previous efforts to estimate the mechanical properties of CNT forests based on their structural morphology was quite successful [46] and gives promise that similar algorithms may enhance the performance predictability of each individual fiber. These endeavors may lead to more sensitive CNT forest sensors and expanded application domains.

## 5. Conclusions

Through combined physical and numerical experiments, we have demonstrated the significant piezoresistive response of CNT-coated microfibers to diametric compression on the order of micrometers. In our test configuration, the electrical resistance was measured across a planar electrode array in an orientation that is parallel to the long axis of the fiber. The magnitude of the resistance change was large (>100 Ohms) and required no signal amplification or filtering. Gauge factors greater than 3.5 are demonstrated, with a resistance sensitivity of up to 35% per micron of compression. Furthermore, the electrical resistance was recoverable upon unloading, with some hysteresis observed.

Electromechanical simulation suggests that CNT–electrode contacts are the mechanism responsible for the largest resistance change. Relatively few CNT free ends contact the measurement electrodes, establishing a bottleneck for charge transfer. As new contact points are established, the overall resistance of the CNT forest electrical network scales inversely. Conversely, many thousands of CNT–CNT contacts were established within the CNT forest prior to compression. The CNT–CNT contacts approximately double during the simulated compression; however, their combined influence on the overall resistance is of an order of magnitude less significant than that of the contributions of CNT–electrode contacts, that increase by tens to hundreds of times.

The experiments and simulation also demonstrated that the idealized cylindrical coating of CNT forests around the fiber substrates becomes increasingly more irregular as the length of CNTs increase. The CNT forests tend to form densified bands around the perimeter of the cylindrical substrate. In the context of using the CNT-forest-coated fiber for mechanical sensing, the irregularities in CNT forest geometry are expected to decrease consistency with respect to both contact area and deformation mechanics. In our diametric compression testing, regions of obvious inhomogeneity were avoided to mitigate these irregularities.

Based on our results, we hypothesize that a small-diameter glass fiber with a short CNT forest coating provides the highest sensitivity with respect to both contact resistance and internal CNT–CNT junction resistance. Although the coating volume is more regular and homogenous for short CNT forests, small asperities at the tips of the CNT forests lead to significant fiber-to-fiber variations. The observed fiber-to-fiber variation is further evidence of the importance of contact resistance to the overall piezoresistive sensitivity. Efforts to further enhance the sensitivity of CNT-forest based sensors include further decreasing the areal density of CNTs to decrease the CNT–electrode contact area and quantity of CNT–CNT junctions in the absence of loading. Smaller-diameter fiber substrates may also increase the sensitivity by similar mechanisms.

## Figures and Tables

**Figure 1 sensors-23-05190-f001:**
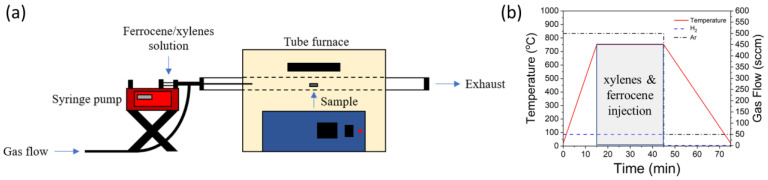
Floating catalyst chemical vapor deposition of CNT forests. (**a**) Schematic of floating catalyst CVD setup. (**b**) CNT forest synthesis conditions.

**Figure 2 sensors-23-05190-f002:**
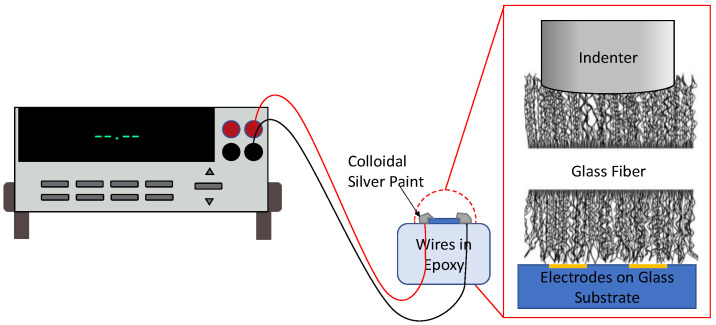
Schematic of the electromechanical test setup. The interdigitated electrode substrate resided with a nanoindenter. A 100 μm flat tip diametrically compressed a CNT-forest-coated fiber onto the patterned substrate while the electrical resistance was measured using a digital multimeter.

**Figure 3 sensors-23-05190-f003:**
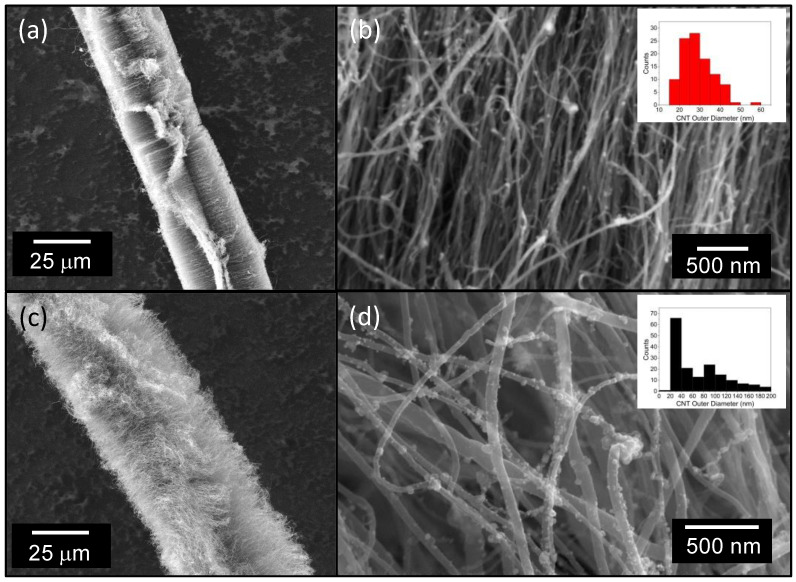
SEM micrographs of CNT-forest-coated glass fibers. (**a**,**b**) Alumina-coated glass fibers. (**c**,**d**) As-received glass fiber substrates. Inset images in (**b**,**d**) show histograms of CNT outer diameters measured by SEM.

**Figure 4 sensors-23-05190-f004:**
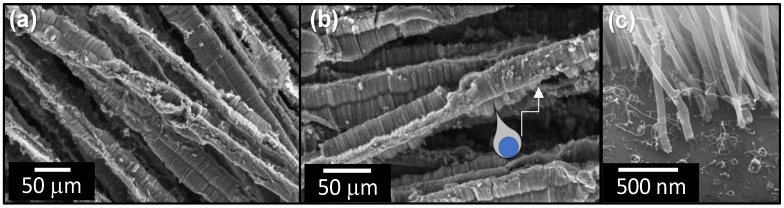
SEM images of CNT-forest-coated microfibers. (**a**) Large-diameter CNT forests exhibit CNT banding instead of a continuous conformal coating. While most fibers exhibit two or more preferred bands, (**b**) some CNT forests collapse into a single band. The schematic in (**b**) demonstrates the position of the fiber (blue) relative to the CNT forest. (**c**) High magnification image shows catalyst nanoparticles on the glass fiber substrate and CNTs separated from the catalyst.

**Figure 5 sensors-23-05190-f005:**
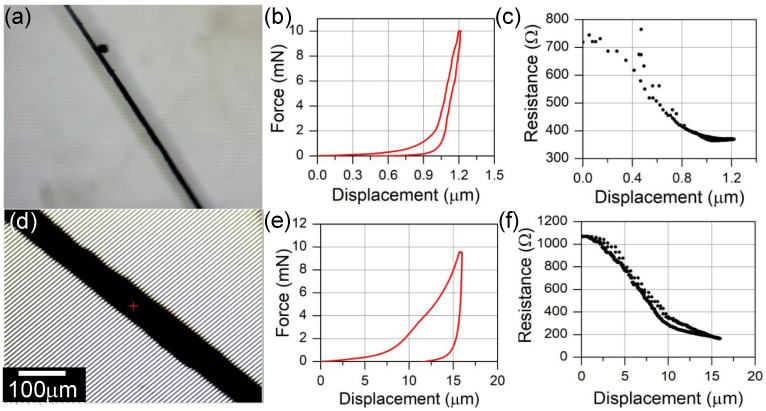
Electromechanical indentation data from diametrically compressed CNT-forest-coated glass microfibers. (**a**) Optical image of CNT-coated fiber tested in (**b**,**c**). (**b**) Mechanical and (**c**) simultaneous electrical data from a 12 μm diameter CNT forest grown from an alumina-coated glass fiber. (**d**) Optical image of CNT-coated fiber tested in (**e**,**f**). (**e**) Mechanical and (**f**) simultaneous electrical data from a 60 μm diameter CNT forest grown from an alumina-coated glass fiber.

**Figure 6 sensors-23-05190-f006:**
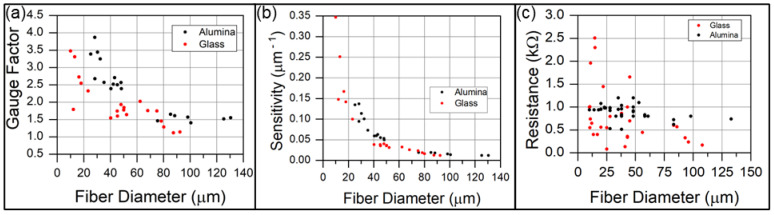
Sensitivity of CNT-forest-coated glass fibers. (**a**) Gauge factor and (**b**) sensitivity of CNT forests grown on alumina-coated and as-received glass fibers. The gauge factor is computed with respect to diametric compression. (**c**) The initial resistance of CNT-coated fibers is inversely related to the fiber diameter.

**Figure 7 sensors-23-05190-f007:**
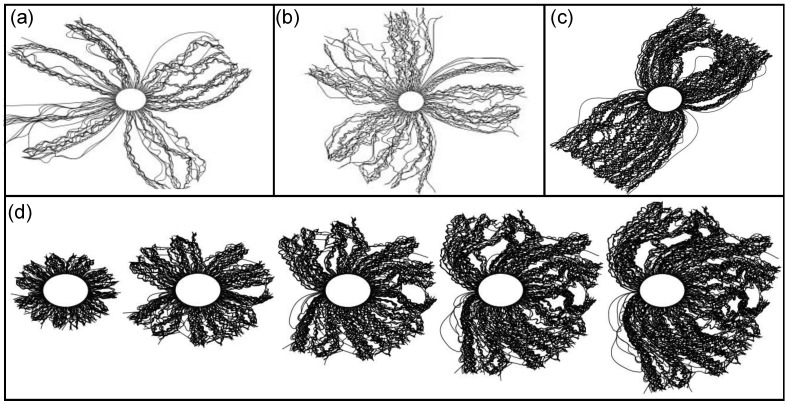
Simulated CNT morphology as a function of synthesis attributes after 600 growth time steps. The attribute combinations include (**a**) 3 × 10^9^ CNT/cm^2^, 40 nm O.D., 34 nm I.D. CNT diameter, 0.1 nN force to break CNT–CNT bonds, (**b**) 3 × 10^9^ CNT/cm^2^, 40 nm O.D., 20 nm I.D. CNT diameter, 1 nN force to break CNT–CNT bonds, (**c**) 3 × 10^10^ CNT/cm^2^, 5 nm CNT diameter, 10 nN force to break CNT–CNT bonds. (**d**) A time evolution shows the formation of a single band using the attributes of 3 × 10^10^ CNT/cm^2^, 5 nm CNT diameter, 1 nN force to break CNT–CNT bonds. Note that the formation of different band structures is stochastic in nature. One set of synthesis attributes will not ensure one specific type of band structure.

**Figure 8 sensors-23-05190-f008:**
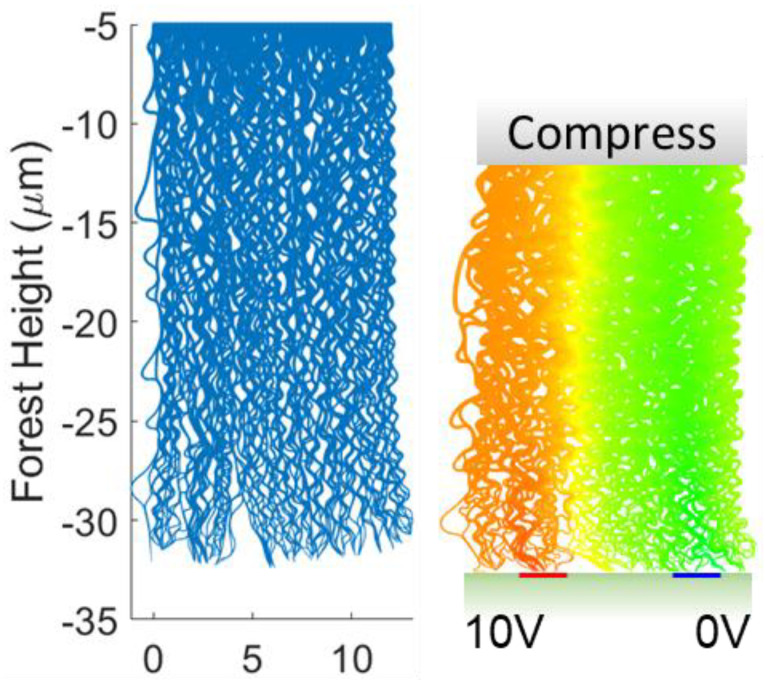
Schematic of a simulated CNT forest for compression (**left**) and the electromechanical compression of the same forest (**right**). The colors within the compressed forest represent voltage.

**Figure 9 sensors-23-05190-f009:**
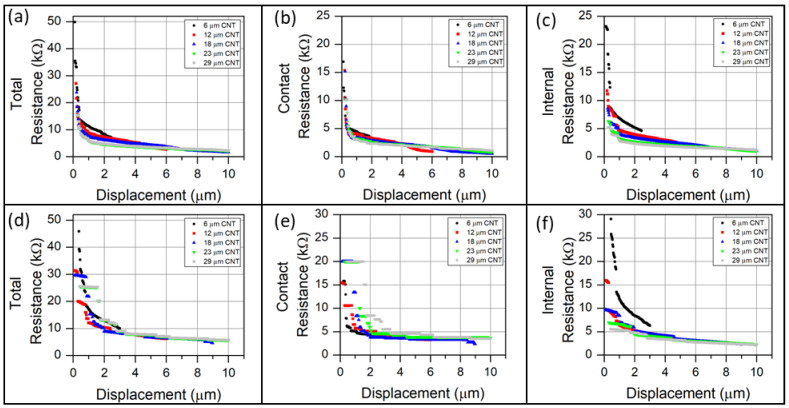
Simulated electromechanical response of compressed CNT forests as a function of compressive displacement. Panels (**a**–**c**) represent simulations with 10 nm outer diameters, and areal density 3 × 10^10^ CNT/cm^2^, to simulate the CNT forests synthesized from alumina-coated fibers. Panels (**d**–**f**) represent CNT forests in which the CNT outer diameter is 40 nm and CNT areal density is 3 × 10^9^ CNT/cm^2^ to simulate the CNT forests synthesized from as-received glass fibers. The legend represents the height of the CNT forest before compression. Panels (**a**,**d**) display the total electrical resistance through the CNT forests between the measurement electrodes. Panels (**b**,**e**) display the electrical contact resistance established between CNTs and the electrode surfaces. Panels (**c**,**f**) display the intrinsic internal resistance within the CNT forest obtained by setting the voltage of nodes in contact with an electrode equal to the voltage of the electrode.

**Figure 10 sensors-23-05190-f010:**
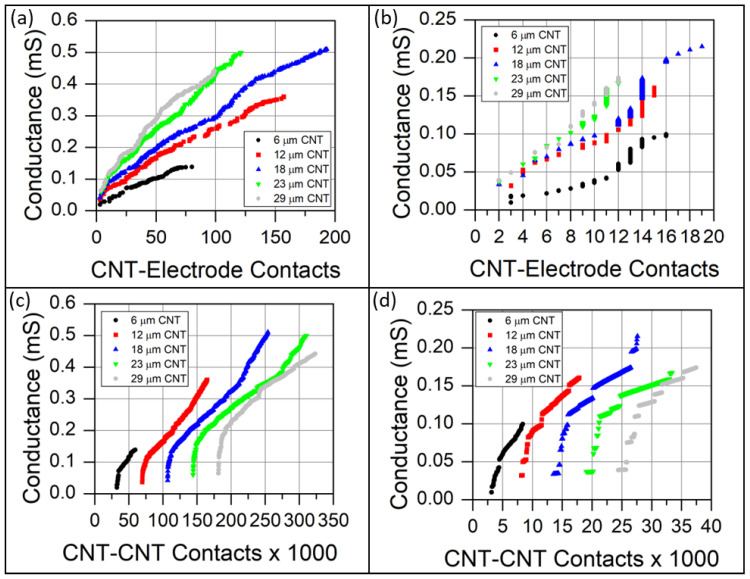
Simulated CNT forest conductance as a function of the quantity of (**a**,**b**) CNT–electrode contacts, and (**c**,**d**) CNT–CNT contacts. Panels (**a**,**c**) represent CNT forests in which the CNT outer diameter is 10 nm and CNT areal density is 3 × 10^10^ CNT/cm^2^ to simulate the CNT forests synthesized from alumina-coated fibers. Panels (**b**,**d**) represent CNT forests in which the CNT outer diameter is 40 nm and CNT areal density is 3 × 10^9^ CNT/cm^2^ to simulate the CNT forests synthesized from as-received glass fibers. The legend represents the height of the CNT forest before compression.

**Figure 11 sensors-23-05190-f011:**
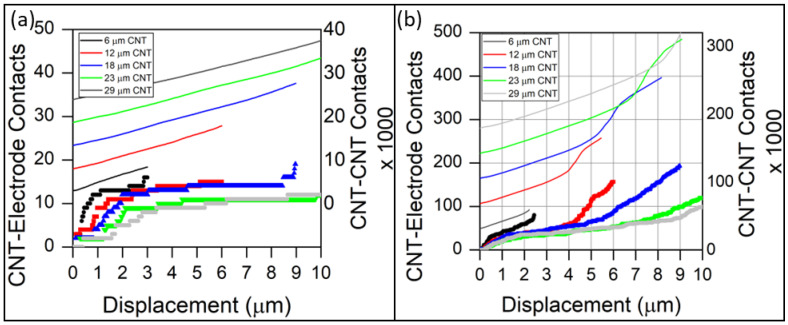
The quantity of CNT–electrode and CNT–CNT contacts as a function of compressive displacement for (**a**) 40 nm O.D., 3 × 10^9^ CNT/cm^2^ CNT forest and (**b**) 10 nm O.D., 3 × 10^10^ CNT/cm^2^ CNT forest. The discretely plotted points represent CNT–electrode contacts, while the continuous lines represent CNT–CNT contacts.

## Data Availability

The data presented in this study are available on request from the corresponding author.

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
