# Peer review of "Investigating the Electromechanical Sensitivity of Carbon-Nanotube-Coated Microfibers"

_sensors, 2023, doi:10.3390/s23115190_

Round 1

Reviewer 1 Report

1.       Add a schematic diagram of the CVD synthesis process.

2.       Also include a Temperature (Y-axis) and Time (X-axis) diagram, mention the gases with a flow rate in proper places

3.       “The CNTs grown on the alumina layer are multi-walled CNTs with a typical outer diameter of approximately 5 – 30 nm (Figure 1b). The CNT outer diameter grown from the as-received fibre are much larger, ranging from 30 – 60 nm (Figure 1d).”- make histograms

4.       Many claims have been made in Figure 2. However, most of them could not be verified in Figure 2. Thus, the authors must replace the current one (figure 2) with a more explicit one and indicate the claims by pointing within Figure 2.

5.       A Schematic of the Electromechanical Experimental setup is required along with the original one (where the experiment was performed).

6.       “Unloading follows a similar slope, with a hysteresis that is commonly observed during compression of flat CNT forests”- add references.

7.       “The data shown in Figure 3 represent cases in which the electrical resistance was restored to initial levels upon removal of the compressive load. In other tests, the electrical resistance recovered partially.”-Why? (The explanation given in the paper is not satisfactory)

8.      In Figure 4c. “Though the data is significantly scattered, a general trend of decreasing initial resistance with increasing diameter is observed.”- vague statement resulting from poor analysis.

9.       Since there are many discrepancies in the experimental section, thus simulation section becomes redundant till the corrections in the experimental section are made. 

It required moderate editing of the English language

Reviewer 3 Report

This article is of high interest for the journal, with some great contribution from experiments and simulation.

I do have some questions mainly related to materials preparation and the CNT characteristic and properties.

Lines 78-81: more details should be provided on the ALD experimental conditions (or clearly refer to a previous article).

Do the authors have any indication regarding the structural integrity (Raman spectroscopy data for instance) in relation with glass fiber substrate (as-received or alumina-coated)?

Figure 4 seems to show some data about fiber diameter size distribution, indeed more information about this (or CNT length), as well as the number of compression tests run, would be useful to assess the reproducibility of the results.

How was the red line in Figure 4c calculated? Fitting is hardly convincing. The authors mention initial resistance in relation with fiber diameter although the Y and X axes are initial resistance and displacement, respectively (X axis is not fiber diameter)?

Lines 254-255: where does the inner diameter/outer diameter ratio of 85% come from, is it backed up by the literature or experimental results? For as-received glass fibers that would correspond to 9 walls CNTs with 34 nm inner diameter which seems high (for alumina-coated fiber, these dimensions would correspond to double walled CNTs with inner diameter of 8.5 nm which is also unusually large for double walled CNTs)

Reviewer 4 Report

sensors-2352775

In the manuscript entitled “Investigating the Electromechanical Sensitivity of Carbon Nanotube Coated Microfibers”, the authors report the electromechanical properties by optimizing the morphology of CNTs. The manuscript deals with an interesting topic for the scientific community, offering some improvements with respect to the related literature. Nevertheless, the article contains shortcomings and inaccuracies, the correction of which can significantly improve the quality of the article.

1-     The background literature review is very brief in the introduction and the aim of the manuscript should be presented with more details at the end of the introduction.

2-     You mentioned high durability as an important factor in choosing CNTs. This claim needs explanation. Or at least add a related reference to the manuscript that compares the durability of CNT with other materials.

3-     In investigating the morphology of forest CNT the number of tubes or fibers per area is an important factor. Discuss this topic as well or refer to the article below. (DOI: 10.3390/ma15041383)

4-     Mention the thickness of the catalyst used in the CNT growth process. The effect of thickness on morphology should be discussed.

5-     The temperature of the growth process is mentioned, but the reactor pressure is not mentioned.

6-     Change 100oC to 100 °C. Apply this change to all text.

7-     What was the reason for choosing 10 mN for a maximum load? And 15s for loading and unloading duration?

8-     Add the specifications of the scanning electron microscope.

9-     This sentence needs more explanation or related reference: “Larger diameter CNTs have a much greater bending stiffness compared to small diameter CNTs, as bending stiffness scales with diameter to the fourth power

10-  It is necessary to provide more explanation about the forces between the tubes in CNT forest-like.

11-  The scale bar in Figure 3d is unclear.

12-  It is highly recommended to add a conclusion section to the article.

N/A

Round 2

Reviewer 1 Report

1. Histograms are not convincing. In fig 3d the histogram shows over 60 CNTs with OD of 30 nm. Please recheck.

2. "two bands originating radially from the fiber substrate."- Provide a clear high resolution SEM/FESEM picture.

3. In Fig 4C the data is so scattered that it is not possible to find any general trend. 

correcting grammatical errors, fixing sentence structure issues, and rephrasing sentences or paragraphs to improve their readability and flow

Reviewer 2 Report

From last revision:

Major comments:

11) Thank you. The introduction is really improved. 

22) Thank you for the additional information. I am still wondering if you obtain additional CNTs down the length of the tube that are not anchored to the fibers.

33) Is this procedure published somewhere? Can you cite it?

44) SEM cannot differentiate an individual CNT from a small bundle of CNTs. I also cannot view comment R3 as it was sent to Reviewer 1.

55)      Thank you for clarifying.

66)      This makes much more sense.

 Minor comments:

Thank you for addressing this comments.

New Comment:

I’m sure this will be addressed copy editing, but line 103 has a figure reference error.

Reviewer 3 Report

Thanks for the revised and improved version however there is one comment (R30) where I am not convinced by the authors answer.

Here is the original comment:

Lines 254-255: where does the inner diameter/outer diameter ratio of 85% come from, is it backed up by the literature or experimental results? […]

And here is the authors answer:

R30) We thank the reviewer for the comment. The inner to outer diameter ratio of 85% is an approximation based on observations from previous floating catalyst CNT syntheses. The text below and reference have been added to the revised manuscript.
The inner to outer diameter ratio was an assumption based on observations from prior floating catalyst CNT syntheses [57-58]”

In the cited references there is no data nor discussion on inner diameter nor on "inner to outer diameter ratio". The only related information comes from Figure 1 in ref. 57 (TEM showing 1 nanotube) and Figure 3e in ref. 58 (TEM showing 1 nanotube).

First, it is not reasonable nor acceptable to support this ratio with only TEM data about 2 carbon nanotubes (1 from each reference where inner diameter is not even mentioned). Second, it is hard to measure anything from TEM in ref. 57 but on the one nanotube shown in ref. 58 (without any information on representativity, dispersion in diameters and so on) the aforementioned ratio is about 34% (inner diameter ≈ 13 nm, outer diameter ≈ 38 nm) which is realistic and quite far from the 85% value claimed by the authors.

So the original question remains, where does this 85% value come from?

Other authors report the inner to outer diameter of vertically aligned CNTs ranging from 65% (external diameter of 6 nm) to 45% (external diameter of 10 nm). Also note that this ratio decreases with increasing external diameter so it is expected to be even lower for CNT with external diameter of 40 nm (in agreement with the 34% ratio found from ref. 58… for a single CNT!).

To sum up, the assumption of inner to outer diameter of 85% used for simulations remains highly dubious, it is contradicted by the new references added in the manuscript and still needs to be justified. I believe this matter needs to be elucidated before publication otherwise it would severely undermine the authors manuscript.

Reviewer 4 Report

N/A

Minor editing of English language required.

Author Response

No new comments were provided by this reviewer.

Round 3

Reviewer 1 Report

I appreciate the improvements made to enhance the paper's quality. However, it is important to address the concern regarding the use of excessive self-citations, as it is generally regarded as an unfavorable practice.

Author Response

We thank you for the comment. We removed several conference citations in the previous draft. We feel that the citation count is appropriate to reflect our unique contributions in the areas of CNT forest simulation, CNT forest piezoresistance, and CNT forest mechanics.

Reviewer 3 Report

I would like to thank the authors for their extensive revision as they took very seriously my comment about inner to outer diameter.

However I am still not entirely satisfied as the references indicated in the manuscript (54 to 57) are not relevant for the inner-outer diameter ratio which is not mentioned in these articles (nor is there any data for inner diameter). The authors should look for more relevant publications adressing this in much more details, see for instance this work: 10.3390/nano12142338

Author Response

We thank the reviewer for the comments. We have included additional references that highlight the CNT diameter via TEM. These include:

  1. Lim, Y.D.; Avramchuck, A.V.; Grapov, D.; Tan, C.W.; Tay, B.K.; Aditya, S.; Labunov, V. Enhanced Carbon Nanotubes Growth Using Nickel/Ferrocene-Hybridized Catalyst. ACS Omega 2017, 2, 6063–6071, doi:10.1021/acsomega.7b00858.
  2. Cheng, J.; Zou, X.P.; Zhu, G.; Wang, M.F.; Su, Y.; Yang, G.Q.; Lü, X.M. Synthesis of Iron-Filled Carbon Nanotubes with a Great Excess of Ferrocene and Their Magnetic Properties. Solid State Commun 2009, 149, 1619–1622, doi:10.1016/J.SSC.2009.06.037.
  3. Wei, Y.; Wang, W.-Q.; Hu, Y.-T.; -, al; Fu, C.C.; Huang, H.Y.; Zhang, J.Y. One-Step Path to Highly Crystalline Multi-Walled Carbon Nanotubes with Large Inner Diameters. IOP Conf Ser Mater Sci Eng 2019, 479, 012112, doi:10.1088/1757-899X/479/1/012112.